

# Historical Nitrogen Fertilizer Use in Agricultural Ecosystem of the Continental United States during 1850-2015: Application rate, Timing, and Fertilizer Types

Peiyu Cao[1], Chaoqun Lu[1], and Zhen Yu[1]

[1]Department of Ecology, Evolution, and Organismal Biology, Iowa State University, Ames, Iowa, USA

*Correspondence to*: Chaoqun Lu (clu@iastate.edu)

**Abstract.** Tremendous amount of anthropogenic nitrogen (N) fertilizer has been applied to agricultural lands to promote the crop production in the United States since the 1850s. However, inappropriate N management practices caused numerous ecological and environmental problems which are difficult to quantify due to paucity of historically spatially explicit fertilizer

use maps. Understanding and assessing N fertilizer management history could provide essential implications for enhancing N use efficiency (NUE) and reducing N loss. In this study, we therefore developed long-term gridded maps depicting crop-specific N fertilizer use rate, timing, and fraction of ammonium N ($NH_4^+$-N) and nitrate N ($NO_3^-$-N) across the contiguous U.S at a resolution of 5 km × 5 km during 1850-2015. We found that N use rates of the U.S. increased from 0.28 g N $m^{-2}$ $yr^{-1}$ in 1940 to 9.54 g N $m^{-2}$ $yr^{-1}$ in 2015. Geospatial analysis revealed that the hotspots of N fertilizer use have shifted from the

southeastern and eastern U.S. to the Midwest and the Great Plains during the past century. Specifically, corn of the Corn Belt region received the most intensive N input in spring, followed by large N application amount in fall, implying a high N loss risk in this region. Moreover, spatial-temporal fraction of $NH_4^+$-N and $NO_3^-$-N varied largely among regions. Generally, farmers have increasingly favored $NH_4^+$-N form fertilizers over $NO_3^-$-N fertilizers since the 1940s. The N fertilizer use data developed in this study could serve as an essential input for modeling communities to fully assess the N addition impacts, and

improve N management to alleviate environmental problems. Datasets available at https://doi.pangaea.de/10.1594/PANGAEA.883585.

**Keywords.** The United States, Cropland, Historical nitrogen fertilizer use, Application timing, $NH_4^+$-N and $NO_3^-$-N fertilizer use

## 1 Introduction

The development of the Haber-Bosch process in the early 1900s led to the massive production of relatively cheap nitrogen (N) fertilizer that boosted crop yields (Erisman et al., 2008; Follett et al., 2010). In the United States, N input from fertilizer increased from less than 1 Tg N per year before 1950 to more than 11 Tg N per year by the beginning of 21 century to increase food production (Ruddy et al., 2006). According to Stewart et al. (2005)'s estimation, N fertilizer was responsible for 26% of



six major non-leguminous crops production increase in the U.S. Zhang et al. (2015) estimated that nitrogen use efficiency (NUE) in the U.S. increased by approximately 30% in the last 40 years, which can be attributed to adopting crop varieties, increasing irrigation, and improving nutrient management (Van Grinsven et al., 2015). However, nearly half of N fertilizer input was not utilized by crops (Smil, 1999; Cassman et al., 2002; Tilman et al., 2002; Ciampitti et al., 2014), and a significant amount of N applied was lost to environment via nitrification, denitrification, leaching, and volatilization. It has caused

numerous environmental and ecological problems, such as greenhouse gas emissions, eutrophication, soil acidification, and biodiversity reduction (Mcisaac et al., 2001; Galloway et al., 2003; Bowman et al., 2008).

It is estimated that agricultural production must be doubled by 2050 to meet the demands of growing human population (Alexandratos and Bruinsma, 2012), under which the NUE in the U.S. needs to be increased by 10% during 2010 to 2050 (Zhang et al., 2015). Nonetheless, there is limited room for yield increment under such a high level of N input in most regions

of the U.S. because of diminishing returns, that is, decreasing yield increment from increasing N fertilizer use (Tilman et al., 2002; Ray et al., 2013). Potential rise of NUE may be achievable by improving N management practices, such as adopting appropriate application timing and N fertilizer forms. In contrast, inappropriate configuration of fertilizer use (e.g. rate, timing, fraction of $NH_4^+$-N and $NO_3^-$-N) could increase N losses. For example, excess N input can significantly increase the $NO_3^-$-N leaching to the aquatic systems through drainage (Jaynes et al., 2001), and raise the emission of $N_2O$ (Davidson, 2009; Millar

et al., 2010; Hoben et al., 2011). Moreover, fall application, which is known to have some negative effects because of the time-gap between nutrient supply and the demand of plants, is still popular in the Midwest due to favorable weather and soil conditions, available labor, and lower fertilizer prices (Dinnes et al., 2002). Such negative impacts including higher potential of N losses (e.g. Ammonia volatilization and N2O emissions) can dramatically reduce NUE (Hao et al., 2001; Randall et al., 2003). Agricultural N losses are also related to the forms of fertilizer applied. For example, ammonium form ($NH_4^+$-N)

fertilizers such as urea promotes rapid volatilization (Vlek et al., 1979; Keller and Mengel, 1986; He et al., 1999), while Nitrate form ($NO_3^-$-N) fertilizers are the major contributors of N leaching (Dinnes et al., 2002). In addition, both Ammonium and Nitrate fertilizers can contribute to N2O emission through nitrification and denitrification under different soil temperature and moisture conditions (Azam et al., 2002; Bouwman et al., 2002; Tenuta et al., 2003; Venterea et al., 2008; Snyder et al., 2009). To better manage the use of N fertilizer and minimize the negative impacts, it is important to examine the historical N use

within spatial and temporal contexts. Ruddy et al. (2006) derived annual N fertilizer input data in the U.S. at county-level using annual commercial fertilizer consumption report in each state provided by the Association of American Plant Food Control Officials (AAPFCO) and the enhanced 1992 National Land Cover Data (NLCDe92) (Nakagaki and Wolock, 2005). Nevertheless, their study was unable to depict cross-crop divergence in N fertilizer use, which is critical for identifying the areas of N deficiency and the hotspots of overfertilization due to different response of crops to N input (Stewart et al., 2005;

Deryng et al., 2011; Mueller et al., 2012). Moreover, Ruddy et al. (2006) only covers N fertilizer use in a short period from 1982 to 2001 based on one-phase static land cover map. A long-term N application history is important for both field investigators and modeling community to comprehensively examine the cumulative impacts of fertilizer uses (Alexander and Smith., 1990; Van Grinsven et al., 2015). Spatially, cropland area change is also an important factor determining the





distribution of agricultural N input. Additionally, the spatiotemporal maps depicting both application timing and fertilizer

forms are essential for accurate assessment of N losses because of the significant interactions between these two (Harrison and

Webb, 2001). Nitrogen can be overwhelmingly lost through ammonia volatilization within a few days after application if the

conditions were suitable (Burch and Fox, 1989; Jarvis and Pain, 1990). For example, Sommer et al. (2004) reported that half

of the total $NH_3$ could be lost from urea in 2-7 days after application. In addition, nitrate fertilizer applied in fall and spring is

vulnerable to leaching by following heavy rainfall due to its high mobility. However, there is still a lack of data for describing

long-term spatially explicit agricultural N management practices across contiguous U.S.

To facilitate earth system modeling and inventory-based studies, we developed a spatially explicit time-series data set to

describe agricultural synthetic N fertilizer input (rate, timing, and fraction of $NH_4^+$-N and $NO_3^-$-N) in the contiguous U.S. at a

resolution of 5 km × 5 km during 1850 to 2015. In this study, we aim at: 1) quantifying commercial N fertilizer application

rate in agricultural land across the U.S. and identifying the historical hotspots of N fertilizer uses; 2) examining the geospatial

patterns of N fertilizer application timing nationwide; and 3) examining the spatial and temporal variations in proportions of

$NH_4^+$-N and $NO_3^-$-N throughout N fertilizer use history across agricultural regions of the U.S.

## 2 Method

We generated annual state-level crop-specific commercial nitrogen (N) use rate during 1850 to 2015 through calculating and

gap-filling national total N fertilizer consumption, national crop-specific N fertilizer average use rate, and state-level N

fertilizer average use rate from multiple sources. We split N use rate generated above into four application timings according

to the state-level crop-specific survey data in the latest years. We further calculated $NH_4^+$-N and $NO_3^-$-N use rate of the four

timings based on state-level fraction of $NH_4^+$-N and $NO_3^-$-N estimated from 11 major single N fertilizer types and their

preferred application timings. We spatialized N fertilizer use records generated above (including rate, timing, and $NH_4^+$-N and

$NO_3^-$-N use rate) to gridded maps based on 1 km × 1 km historical land cover data of the contiguous U.S. developed by Yu

and Lu (2017) (Fig. 1).

### 2.1 Historical N fertilizer use rate reconstruction

*Estimating national commercial farm-used N fertilizer consumption*. By harmonizing the annual national commercial farm-

used N consumption from Mehring et al. (1957) for 1850-1951, USDA (1971) for 1952-1959, and USDA-ERS (2013) for

1960-2011, we obtained the N consumption record of the contiguous U.S. from 1850 to 2011.

*Estimating state-level N fertilizer use rate.* By integrating and gap-filling the annual state-level farm-used N fertilizer

consumption from Mehring et al. (1957), USDA (1971), USDA (1977), Brakebill and Grinberg (2017), and national N

consumption generated above, we reconstructed the state-level N fertilizer consumption at the contiguous U.S. from 1850 to

2015. For period before 1930, the state-level N consumptions were unavailable. We assumed that state-level N fertilizer use

followed the trend of the historical national N consumption. Equation (1) was used to retrieve back N fertilizer use of each





state.

$$State\ consumption_i = \frac{National\ consumption_i}{National\ consumption_{i+1}} \times State\ consumption_{i+1} \qquad (1)$$

where *State consumption* refers to the state-level N fertilizer consumption, the *National consumption* refers to the national N fertilizer consumption, and *i* is the years between 1850 to 1930.

We divided the historical N consumption amount by annual cropland area in each state derived from historical land use data

in Yu and Lu (2017) to yield the state-level average N use rates (Supplementary table S1).

***Estimating referenced state-level crop-specific N use rate.*** Using data from Mehring et al. (1957), USDA (1957), Ibach et al. (1964), Ibach and Adams (1967), USDA-ERS (2013), and USDA-NASS (2017) (supplementary table S2), we generated national crop-specific fertilizer use rates for the period of 1850-2015. We focused on nine major crop types nationwide including corn, soybean, winter wheat, spring wheat, cotton, sorghum, rice, barley, and durum wheat. We further gap-filled

the national crop-specific N use rates by using state-level N rates derived above, which served as the reference for gap-filling of the state-level crop-specific data later.

Mehring et al. (1957) provided national total fertilizer consumption (i.e. nitrogen, phosphate, and potassium) of corn, soybean, wheats (winter wheat, spring wheat, and durum wheat in total), cotton, barley, and rice in 1927, 1938, 1942, 1946, and 1950, and specifically reported the N consumptions in 1950. Thus, the ratio of N to total fertilizer in 1950 was calculated and applied

to determine N consumption of the prior four reported years. The N use rate for each crop type was then calculated by dividing the N consumption to planting area of each crop for the period of 1927 to 1950.

The survey datasets after 1950 provided commercial N application rates of nine crops in the croplands that were fertilized. Since there is no information to identify which cropland was fertilized spatially, here, we assumed all croplands in each state were fertilized, and then adjusted the rates by multiplying fertilized cropland percentage with application rate.

Three approaches were adopted to impute the N use rates in missing years based on state-level N use rate derived above. For the period before 1927, the national average crop-specific application rates were unavailable. We used Eq. (1) to retrieve back N fertilizer use rate of each crop.

For the period of 1927 to 2015, the cubic spline interpolation method was used to gap-fill N use rates when data is missing in less than three consecutive years. We assumed the trends of N application rate were relatively smooth during such a short

period. Nonetheless, using interpolation methods may fail to reflect the sharp variations due to changes in fertilizer price and grain demand during a long period (e.g., > 3 years, Fig. 2). Therefore, if the missing data were found in more than three consecutive years, we assumed the crop-specific N use rate followed the inter-annual variation in the state-level rates. We applied the distance-weighted imputation in gap-filling using Equation (2) by assuming the missing data is closer to the rate of the nearest year.


$$Crop\ Rate_{i+k} = \frac{State\ Rate_{i+k} \times Crop\ Rate_i}{State\ Rate_i} \times \frac{k-i}{j-i} + \frac{State\ Rate_{i+k} \times Crop\ Rate_j}{State\ Rate_j} \times \frac{j-k}{j-i} \qquad (2)$$



where *Crop Rate* refers to the average crop-specific N use rate, *State Rate* refers to the State-level N use rate, the years *i* and *j* were the beginning and ending year of the gap, and the year *i+k* was the $k^{th}$ missing year.

***Estimating crop-specific N use rates at state-level.*** The state-level crop-specific N use rates of nine major crops for the period 1954-2015 were derived from the same five data sources (supplementary table S2). N use rates were also adjusted by multiplying fertilized cropland area percentages and use rates.

State-level N use rate of wheat from 1965 to 1989 reported by USDA-ERS (2013) was the weighted average rate of winter wheat, spring wheat, and durum wheat. We calculated the fraction of N consumption of each wheat type in each state to total N consumption of wheats in the year 1990. The fractions were used to estimate N consumption for each wheat type during the period. N use rate of these three wheat types was then calculated by dividing N consumption by planting area of each wheat.

For the period of 1850 to 1953 when state-level N fertilizer use rates of nine crops were unavailable, we gap-filled the missing years using Eq. (1) based on the data in the year 1954 using referenced state-level crop-specific rates. To gap-fill the missing years during 1954-2015, we first built regression models between the referenced crop-specific rates and raw state-level rates of nine crops using quadratic, cubic, exponential, and logarithmic functions. Models that poorly fitted were discarded. The "Best-fit" model was adopted and used to correct the referenced state-level crop-specific N use rate trend, which was then further used in distance-weighted imputation (Eq. 2) or cubic spline interpolation for those cases with missing data over or less than 3 years, respectively. The referenced crop-specific rates were used as a general reference trend when all four models were discarded (e.g. soybean and cotton).

We assume cropland pasture was not fertilized until 1945 due to a lack of area data and low N use (< 1.5% of national total N use in 1942, Mehring et al., 1957). By timing the annual state-level N consumption with the ratio of cropland pasture N consumption to N consumption in each state of the year 1954 (USDA, 1957) for the period 1945-1959, and the year 1964 (Ibach et al., 1967) for the period 1960-2015, we obtained state-level N consumption of cropland pasture from 1945 to 2015. We divided the N consumption by annual cropland pasture area (USDA-NASS, 2017) to generate the state-level cropland pasture N use rate. We further calculated N consumption and crop area of all other crops in each state by subtracting data of nine major crops and cropland pasture from national total. We generated the state-level N use rate for other crops by dividing the N consumption amount by area of "others".

**2.2 Nitrogen application timing**

We adopted single-year timing information for nine major crops in each state (i.e. corn, soybean, winter wheat, spring wheat, cotton, sorghum, rice, barley, and durum wheat). According to USDA-ERS (2016), we grouped state-level crop-specific N fertilizer use into four timings: Fall (previous year), Spring (before planting), At planting, and After planting. ARMS conducted the survey since 1996 and collected the data periodically for each crop. For example, corn producers were surveyed in1996-2001, 2006, and 2010 (Supplementary table S3). Because the collected dates for different crops in each state varied, we adopted the latest survey for all nine crops. The raw data include state-level crop-specific fertilizer use rates at four timings and the percentages of the fertilized cropland in each state. As no spatial information is available to identify where was fertilized, we



assume all cropland was fertilized at a lower application rate by multiplying the reported rate with fertilized cropland percentage in all four application timings (Goebes et al., 2003). We used the fraction of the application rate of each timing to split annual state-level crop-specific N use rate generated in section 2.1 into four timings.

We assume farmers have the same preference in application timings for all crops in each state. We calculated the average application timing fraction based on the fraction of eight crops (excluding winter wheat) generated above for cropland pasture and other crops.

## 165 2.3 Characterizing $NH_4^+$-N and $NO_3^-$-N use rates across states and application timings

***Estimating national consumption of N fertilizers.*** We collected data of national consumptions of 11 major single N fertilizers including Anhydrous Ammonia (AnA), Aqua Ammonia (AqA), Ammonium Nitrate (AN), Ammonium Sulfate (AS), Nitrogen Solution (NS), Sodium Nitrate (SN), Urea, Calcium Nitrate (CN), Diammonium Phosphate (DAP), Monoammonium Phosphate (MAP), and Ammonium Phosphates (APs), since 1900 from Mehring et al. (1957), USDA (1966), USDA-ERS

(2013), and FAO (2017) (supplementary table S4). Among all these 11 N fertilizers, APs refers to the integration of five major forms of Ammonium Phosphate. Before 1960, the consumptions of DAP and MAP were relatively small, which were included in the reported APs. While starting from 1960, the consumptions of these two fertilizers increased sharply and were therefore reported separately. The gaps after 1900, such as the missing years from 1954 to 1959 of CN, were imputed by Eq. (2) based on the national commercial N consumption generated in section 2.1.

Before the wide use of Haber-Bosch process in the U.S., the commercial N fertilizer was mainly extracted from organic products (e.g., Peruvian guano, fish scraps, and dried blood) and mined from mineral deposits (e.g., Chilean saltpeter) (Sheridan, 1979). We assume the fraction of $NH_4^+$-N and $NO_3^-$-N before 1900 remained the same as the level of the year 1900.

***Estimating State consumption of nitrogen fertilizers.*** We collected the state-level records of 11 N fertilizers consumptions from 1946 to 2012 in a 10-year interval from USDA (1966, 1971, and 1977) and AAPFCO (1986, 1996, 2006, and 2012)

(supplementary table S5). The consumptions of CN during 1946-1976, and DAP, MAP during 1966-1976 were unavailable. Thus, we imputed the missing years with the data of the year 1986 using Eq. (1). Since the ratio of a specific fertilizer type consumed in a certain state to entire nation stays relatively stable within one decade (Supplementary figure S1), we used mid-decade ratio to represent the annual ratio in this decade to fill the gaps.

It is well known that applying the liquid fertilizer (NS) and $NO_3^-$-N fertilizer (e.g. SN and CN) in fall and spring increase the

risk of N loss (Randall et al., 2003, 2008). Therefore, preferred application timings vary among different N fertilizers. To characterize the use of N fertilizer types in different timings, we split the consumptions of each N fertilizer type into four timings according to the Agronomy Guide (Supplementary table S6) (Mengel, 2017). Due to the application of nitrification inhibitors, the convenience of applying fertilizer in fall and spring, and the limitation of using equipment, the practical situations may vary. We considered a few practical situations from the survey (Bierman et al., 2012) to adjust the Agronomy

Guide and generate the application ratios of each N fertilizer type for four timings that is closer to actual management practices (Table 1).



After allocating each of the fertilizer type to four timings, we calculated the state consumption of ammonium N ($NH_4^+$-N) and nitrate N ($NO_3^-$-N) at each timing based on the N content and the N form of each fertilizer type listed in table 2 (USDA-NRCS, 2017). Finally, we calculated the fraction of $NH_4^+$-N and $NO_3^-$-N to total N consumption respectively, and then obtained use rate of $NH_4^+$-N and $NO_3^-$-N at each timing for all crops.

## 2.4 Spatializing state-level crop-specific N fertilizer input to gridded maps

For spatial analysis, we downscaled the imputed state-level crop-specific N management data to gridded maps based on 1 km × 1 km historical land cover data of the contiguous U.S. from 1850 to 2015 developed by Yu and Lu (2017). In this paper, for display purpose, we timed the cropland percentage with N use rate in each grid to convert the unit of N use rate from g N per cropland area per year to g N per land area per year, then we resampled the N management maps to a 5 km×5 km resolution with the average fertilizer use rate depicted in each pixel.

To describe the regional difference of N management in the study area, we partitioned the entire study area into seven regions (Fig. 4): the Northwest (NW), the Southwest (SW), the Northern Great Plains (NGP), the Southern Great Plains (SGP), the Midwest (MW), the Southeast (SE), and the Northeast (NE) according to the U.S. Fourth National Climate Assessment (2017).

## 3 Results

### 3.1 Nitrogen fertilizer application rate

N fertilizer consumptions in the U.S. were very low (< 0.4 Tg N yr$^{-1}$) from 1850 to 1940, and then sharply increased to 10.3 Tg N yr$^{-1}$ by 1980, followed by a slower rise to 12 Tg N yr$^{-1}$ by 2015 (Fig.3a). In comparison, the N fertilizer use rate followed the trends of total consumption and increased from less than 0.01 g N m$^{-2}$ yr$^{-1}$ in 1850 to 9.5 g N m$^{-2}$ yr$^{-1}$ in 2015 (Fig. 3a). From 1940 to 1960, moderate rise of application rate was found in almost all crop types with increments ranging from 0.09 g N m$^{-2}$ yr$^{-1}$ in soybean to 5.5 g N m$^{-2}$ yr$^{-1}$ in rice (Fig.3b). The following two decades from 1960 was the period when the most dramatic increase of N fertilizer use occurred. The average increase was 4.1 g N m$^{-2}$ yr$^{-1}$ with the largest increase found in corn (11.2 g N m$^{-2}$ yr$^{-1}$), while cotton received relatively stable amount of N fertilizer during 1960-1980 (Fig. 3b). From 1980 to 2015, minor changes (< 2 g N m$^{-2}$ yr$^{-1}$) of application rate were found in corn, yet large increases were found in rice (7 g N m$^{-2}$ yr$^{-1}$), spring wheat (6.5 g N m$^{-2}$ yr$^{-1}$), and durum wheat (5.2 g N m$^{-2}$ yr$^{-1}$).

Hotspots of the N fertilizer uses shifted from the southeastern and eastern U.S. to the Midwest, the Great Plains, and the Northeast of the U.S. since 1900 (Fig. 4). Application rates of commercial N fertilizer were very low (less than 0.2 g N m$^{-2}$ yr$^{-1}$) across the contiguous U.S. before 1940. Mild increases (1-2 g N m$^{-2}$ yr$^{-1}$) of N fertilizer use were detected along the West Coast and in the southeastern and eastern U.S. in the 1940s. The application rates in these regions showed dramatic increases to above 7 g N m$^{-2}$ yr$^{-1}$ until 1980. In comparison, the Midwest received N fertilizer input for over 10 g N m$^{-2}$ yr$^{-1}$, and the Southern Great Plains, south of the Northern Great Plains, and the Northeast received N fertilizer at a rate of 4-7 g N m$^{-2}$ yr$^{-1}$ on average from 1960 to 1980. Not surprisingly, the most intensive N input were found in the central corn-belt, such as Iowa,



Illinois, Nebraska, and Minnesota, rendering the Midwest the hotspots of N fertilizer use in the U.S. after 2000.

Hotspots of the N fertilizer uses shifted from the southeastern and eastern U.S. to the Midwest, the Great Plains, and the

Northeast of the U.S. since 1900 (Fig. 4). Application rates of commercial N fertilizer were very low (less than 0.2 g N $m^{-2}$ $yr^{-1}$) across the contiguous U.S. before 1940. Mild increases (1-2 g N $m^{-2}$ $yr^{-1}$) of N fertilizer use were detected along the West Coast and in the southeastern and eastern U.S. in the 1940s. The application rates in these regions showed dramatic increases to above 7 g N $m^{-2}$ $yr^{-1}$ until 1980. In comparison, the Midwest received N fertilizer input for over 10 g N $m^{-2}$ $yr^{-1}$, and the Southern Great Plains, south of the Northern Great Plains, and the Northeast received N fertilizer at a rate of 4-7 g N $m^{-2}$ $yr^{-1}$

on average from 1960 to 1980. Not surprisingly, the most intensive N input were found in the central corn-belt, such as Iowa, Illinois, Nebraska, and Minnesota, rendering the Midwest the hotspots of N fertilizer use in the U.S. after 2000.

### 3.2 Application timing

N fertilizer application rate in the four timings varied among regions and time periods. Here we present the seasonal variation of agricultural N input in the year of 2015 as an example (Fig. 5). The contiguous U.S. generally received low N input (< 0.5

g N $m^{-2}$ $yr^{-1}$) in fall, except California, North Dakota, the Midwest, and the Southern Great Plains (2-4 g N $m^{-2}$ $yr^{-1}$). Some of the areas in Washington, eastern Arkansas and southern Louisiana even received fertilizer input more than 4 g N $m^{-2}$ $yr^{-1}$. In comparison, majority of N (> 4 g N $m^{-2}$ $yr^{-1}$) was applied in spring before planting, and the most intensively applied area were the Midwest, the Northern Great Plains, the Northwest, northern Kentucky, and parts of California, Florida, Arkansas and Louisiana (Fig. 5b). Small amount of N (< 1 g N $m^{-2}$ $yr^{-1}$) was applied at planting in most areas (Fig. 5c), though exceptions

were found in the Northern Great Plains, northern California, and Washington (> 3 g N $m^{-2}$ $yr^{-1}$). In comparison, intensive N uses (> 4 g N $m^{-2}$ $yr^{-1}$) after planting were identified in the west of the Midwest, the south of the Northern Great Plains, Idaho, California, and the west and southeast part of the Southeast (Fig. 5d).

High N input in fall prevailed in the Southern Great Plains, and Washington (Supplementary fig. S2a). Besides, relatively high portions of N were also applied in Iowa (31%) and southern Minnesota (32%) in fall. In comparison, N applied in spring

dominated across the continental U.S., especially in the Midwest, the Northwest, east of the Northern Great Plains, Arkansas, and Virginia (> 50%, Supplementary fig. S2b), while N applied at planting was found generally low (< 20%) except in Montana (43%), Wyoming (40%), and New York (49%) (Supplementary fig. S2c). N applied after planting was very high (> 50%) in Ohio, Missouri, West Virginia, Vermont, New Hampshire, and coastal area of the Southeast (Supplementary fig. S2d).

### 3.3 Proportion of $NH_4^+$-N and $NO_3^-$-N in fertilizer use

The fraction of $NH_4^+$-N and $NO_3^-$-N varied greatly among seven regions of the U.S. before the 1940s (Fig. 6). Ammonium Sulfate was the major N fertilizer in the Midwest, the Northwest, and the Southwest, rendering high fraction of $NH_4^+$-N in these regions, while Sodium Nitrate was popular in the Northeast, the Southeast, and the Southern Great Plains, lowering the $NH_4^+$-N fraction in these regions. A wide variety of N fertilizers were adopted from 1940 through 1980 in all regions, among which the high $NH_4^+$-N concentration fertilizers such as Anhydrous Ammonia, Ammonium Nitrate, and N Solution were



favored. Since then, the fraction of $NH_4^+$-N in seven regions remained in high level and relatively stable, however, farmers in these regions shifted from multiple nutrient fertilizers toward single nutrient with high level of N concentration, such as Anhydrous Ammonia, N Solution, and Urea.

The fraction of $NH_4^+$-N to total N use ranged from 0-0.65 in the 1900s to 0.8-0.9 in the 2010s across the U.S. (Fig. 7). $NH_4^+$-N fertilizer accounted for about 70% of the total N fertilizer used in the Northern Great Plains in 1900, while the Northeast
and the Southeast were dominated by $NO_3^-$-N fertilizer which accounted for about 95% of total N fertilizer use. The fractions showed a general downward trend during 1920-1940 except a prominent peak value in 1932. In the following three decades from 1940 to 1970, sharp increases were detected in the Northeast (from 0.21 to 0.75), the Southeast (from 0.09 to 0.75), and the Southern Great Plains (from 0.6 to 0.9). During the period of 1970 to 2015, the fractions of $NH_4^+$-N in all regions were above 0.8, and four of them exceeded 0.9 (Fig. 7).

The proportion of ammonium and nitrate fertilizer uses varied among regions. We took the year of 2015 as an example to address the spatial variation of dominant fertilizer form and its application rate across the U.S. (Fig. 8). The ammonium fertilizer applied in fall mainly concentrated in the central area of contiguous U.S, Washington, and California. with the application rate at 1-3 g N m$^{-2}$ yr$^{-1}$, yet small amount of nitrate fertilizer ($< 0.1$ g N m$^{-2}$ yr$^{-1}$) was applied in fall across the country except some areas in the Southern Great Plains and the Southeast (0.1-0.5 g N m$^{-2}$ yr$^{-1}$). In comparison, ammonium
fertilizer has been intensively used in spring ($> 5$ g N m$^{-2}$ yr$^{-1}$) in the Midwest, the Northern Great Plains, west of the Southeast, and the Northwest, where 0.5-1 g N m$^{-2}$ yr$^{-1}$ of nitrate fertilizer was also applied in the same time period. High-level ammonium fertilizer applied at planting mainly distributed in the Northern Great Plains, Iowa, Wisconsin, Washington, and California with application rate at 1-3 g N m$^{-2}$ yr$^{-1}$. Large amount of ammonium fertilizer uses after planting (above 3 g N m$^{-2}$ yr$^{-1}$) were found in California, Nebraska, the Midwest, the Northern Great Plains, the north of the Southern Great Plains, the west and
southeast of the Southeast, and nitrate fertilizer use rates were also high in these areas, especially along southeast coast (Fig. 8).

## 4 Discussion

### 4.1 Comparison with existing N fertilizer map

In order to examine the validity of our data, we compared our product with other studies. Ruddy et al. (2006) developed a map
showing the spatial pattern of N fertilizer use rate in the contiguous U.S. in 1997 by allocating county-level N consumption data derived from state-N fertilizer sale and agricultural land acreage. The overall pattern of N fertilizer use from our study is similar to Ruddy et al. (2006), indicating the hotspots of N fertilizer use ($>8$ g N m$^{-2}$ yr$^{-1}$) in the Midwest, West and East Coast, and some parts of the Northwest, the Southern Great Plains, Nebraska, California, and Texas (Supplementary fig. S3). Nevertheless, discrepancies were also found in some states, such as California, southern Florida, and eastern North Carolina
($> 8$ g N m$^{-2}$ yr$^{-1}$ in Ruddy et al. (2006) and 2-6 g N m$^{-2}$ yr$^{-1}$ in our study). The higher N use rates showed in Ruddy et al. (2006) amplified fertilizer use in low cropland coverage areas, while our maps removed such "appeared peak" by accounting cropland



density in each grid cell. In comparison, the N use rate of Indiana and Ohio in Ruddy et al. (2006) (4-8 g N m$^{-2}$ yr$^{-1}$) is lower than that in our study (>8 g N m$^{-2}$ yr$^{-1}$). This may be caused by using the fixed cropland map of the year 1992 instead of 1997 in their study, and overlooking cropland area change information over years. In addition, our map provides more details in

spatial heterogeneity by adopting state-level crop-specific N fertilizer application rates in each state and generating spatial maps based on 1-km land use history data (Yu and Lu, 2017). Thus, our map is advantageous in characterizing cross-crop differences in using N fertilizers, especially in the intensively cultivated regions, such as the Midwest, the Great Plains, and south of the Southeast.

We further compared the ratios of N application timing across corn, cotton, spring wheat, and winter wheat nationwide with

the report provided by Wade et al. (2015), in which the conservation practices, such as tillage, nitrogen management, and cover crop, were reported regional and national across the U.S. (supplementary table S7). In our study, the application timing showed that the majority of N applied in spring were found in corn (50%) and spring wheat (43%), in which fall application accounts for the second largest share of annual fertilizer consumption (18% for corn and 22% for spring wheat). In comparison, cotton and winter wheat producers applied 53% and 44% of N after planting, respectively. Overall, the differences between our study

and Wade et al. (2015) range between 0-6%, with the largest difference found in after-planting fertilizer use in cotton (6%). Bierman et al. (2012) reported that corn in Minnesota received 32.5% and 58.8% of N fertilizer in fall and spring of 2009, respectively, which is close to 31.9% (fall) and 59.2% (spring) estimated by our study. These comparisons imply that the N fertilizer application timing across nation and crops were well depicted in our study.

**4.2 Temporal and Spatial change in nitrogen fertilizer use**

Compared to manure, chemical N fertilizer gradually became the major agricultural N input in the U.S. from 1850 (6.67%) to 1930 (83.86%), which however, was still low in total consumption and per unit area rate during this period (Mehring et al., 1957). Although cropland area declined between 1940 and 1970, especially for corn and cotton (Gunjal et al., 1980; Nickerson et al., 2011), the total N consumption at national scale had been increasing due to the rise of N use rate and the widespread adoption of N fertilizer (Beddow, 2012) along the West Coast, the Southeast, the Southern Great Plains, and the Midwest.

With the cropland rose to 383 million acres in 1982 (Nickerson et al., 2011), together with the markedly increase in application rate of all crops except cotton, the major agricultural regions received tremendous amount of N during this period. Driven by the change of grain demand and fertilizer prices, N fertilizer use in the U.S. gradually increased with more fluctuation after 1982. The increase of N fertilizer use in the seven regions can be attributed to different driving factors. For example, the increase of N use in regions dominated by spring wheat and rice, such as north of the Northern Great Plains, west of the

Southeast, and southern Texas, were largely due to the increased fertilizer use rate. In contrast, the increases in the Midwest and south of the Northern Great Plains were mainly due to corn expansion (Yu and Lu, 2017), while the N use rate in corn has been relatively high and stable since the 1980s. It is also worth to mention that, although the corn-belt received a majority of N fertilizer across the U.S. in past 20 years, nearly half of the planting areas under fertilization was the wide cultivation of soybean in this region, which requires little anthropogenic N input (Drinkwater et al., 1998). Along with the rise in N fertilizer



use, the fraction of $NH_4^+$-N to total N use increased since 1900 to present as well. Before the 1960s, the fraction varied greatly among seven regions of the U.S. due to different species of N fertilizers used, which may be determined by fertilizer prices, farmer preference, and cultivated crop types, etc. $NH_4^+$-N fraction remained at high level ($> 80\%$) across the U.S. since the 1970s, which may largely lower the risk of N leaching (Gentry et al., 1998). Nonetheless, it may rise the potential of N loss through ammonia volatilization. In addition, shifting from multiple $NH_4^+$-N fertilizers to single $NH^{4+}$-N fertilizers with high

level of N concentration since the 1960s may also enhance the risks of gaseous N emission due to large amount of N available in soil (Harrison and Webb, 2001).

The winter wheat in the Southern Great Plains and the north of the Northwest received the most of annual ammonium N and nitrate N fertilizer is fall, which contributed to the rise of grain yield (Mahler et al., 1994), while corn in Minnesota, Iowa, and Illinois in the Midwest received over 30% of annual $NH_4^+$-N in fall, implying high potential of N loss in this region (Dinnes

et al., 2002; Parkin et al., 2010). Corn farmers that adopted fall application in the Midwest usually applied nitrification inhibitor with fertilizer, however, sudden $NH_4^+$-N input exceeding plant demands may cause tremendous ammonia volatilization (Sommer et al., 2004). Soybean and winter wheat in the Midwest, north of the Southeast, and east of the Northern Great Plains received more than 65% of N in fall and after planting respectively. However, due to the widely cultivated corn with two-fold higher N application rates than other crops, these regions received most ammonium N, as well as nitrate N, in spring. According

to the USDA, "spring application" represents N used at approximately one month before planting. Thus a one-month earlier of N use before the onset of plant growth along with the largest fertilizer input in a year may cause large amount of N losses, such as leaching of soluble nitrate triggered by intense rainfall in spring, contributing to hypoxia in the Gulf of Mexico (Goolsby et al., 2001). In comparison, fertilizer uses for spring wheat and barley were only found at-planting in Montana and North Dakota in form of ammonium N. Moreover, cotton farmers along the southeast coast and in western Texas preferred to

apply both ammonium- and nitrate-form fertilizer after planting, which is the case of corn and cotton farmers' preference in Ohio and Nebraska, and the case of winter wheat in the Southern Great Plains.

### 4.3 Uncertainty and future research needs

The uncertainties of this database are mainly from the following aspects: 1) Although N fertilizer use in the 9 major crops and cropland pasture together accounts for 84% of national total fertilizer consumption, a relatively small portion of fertilizer goes

to other crops. We grouped those crops (e.g. oilseeds, sugar crops, vegetables, fruits, other grains) into a category of "others" and equally assigned the rest N fertilizer amount (rate, timing, and fraction of $NH_4^+$-N and $NO_3^-$-N) to all the crops within others. This may bias the estimated N fertilizer use of some crop types. 2) Due to the paucity of finer resolution information, state-level N management data (rate, timing, and fraction of $NH_4^+$-N and $NO_3^-$-N) were obtained to characterize the way in which farmers use fertilizer in maximizing crop production. Although this is the best data we can obtain for national coverage

and centurial duration, it lacked in-state details. 3) Timing information of the crop-specific N application derived from the latest survey years was assumed unchanged over time due to the lack of inter-annual survey data. This assumption may cause underestimation of fall application before 2000s, as urea and N solution-forms of N, which are suitable for spring application,

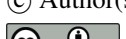



have been increasingly used to replace fall-applied anhydrous ammonia since 1960s (Randall et al., 2008). 4) To separate the amount of $NH_4^+$-N and $NO_3^-$-N in fertilizer use, we adopted the N form fraction of 11 major single N fertilizer types, which

accounted for 85% of the N consumption in the U.S. after the 1980s, and assumed the $NH_4^+$-N: $NO_3^-$-N ratio is 1:1 for the rest fertilizer types (Supplementary fig. S3). However, mixed N fertilizers were more favored before single N fertilizers gain in popularity (Sheridan, 1979). The temporal change of $NH_4^+$-N fraction in mixed N fertilizer before the 1980s may be biased in estimations. For example, ammonium N accounted for only 2% of mixed fertilizer N in 1900, but it was favored since 1925 and gradually rose to 72% of mixed N fertilizer by 1944 (Mehring et al., 1946). 5) $NH_4^+$-N fraction was assumed to be constant

across crop types in each state in a year. This may cause biases because farmers may apply different type of N fertilizers to different crops. For example, $NH_4^+$-N was favored in rice paddy due to higher oxidicability and tendency of N loss via denitrification from $NO_3^-$-N, implying a higher $NH_4^+$-N fraction in the N fertilizer of rice (Norman et al., 2003). Therefore, a finer-scale spatial survey of crop-specific N fertilizer use (e.g. county-level), annual application timing data, and development of crop-specific fraction of $NH_4^+$-N and $NO_3^-$-N data will be beneficial for further improving characterization of geospatial

and temporal patterns of N fertilizer management in the U.S.

**5 Data availability**

The N fertilizer use dataset is publicly available via PANGAEA at https://doi.pangaea.de/10.1594/PANGAEA.883585 (Cao et al., 2017).

**6 Conclusion**

Nitrogen fertilizer management (e.g. N fertilizer use rate, application timing, and fraction of $NH_4^+$-N and $NO_3^-$-N) is a critical component of agricultural practice that significantly promotes crop yield. The dataset developed in this study enables us to explore the spatiotemporal pattern of N fertilizer management across the U.S. N fertilizer consumption, as well as N fertilizer use rate increased tremendously from 1850 to 2015, but the magnitude varied among crop types. Meanwhile, hotspots of N fertilizer uses shifted from the southeastern and eastern U.S. to the Midwest, the Great Plains, and the Northeast of the U.S.

since 1900. In addition, the majority of N fertilizer was applied in spring, approximately one month before planting. Moreover, considerable amount of N fertilizer was applied in fall in previous year, which implies high risk of gaseous N emission and N leaching. The fraction of $NH_4^+$-N varied greatly among seven regions of the U.S. before the 1960s, while the $NH_4^+$-N gained the popularity and dominated the N fertilizer use after the 1970s, which reduce the potential of N loading while increasing ammonia volatilization risk. Appropriate configuration of N fertilizer use according to the fertilizer demands spatially should

be encouraged to improve nitrogen use efficiency and thus reduce environmental and ecological problems.

**Competing interests.** The authors declare that they have no conflict of interest.



**Acknowledgements**

This work was supported by seed grant from Iowa Nutrient Research Center and new faculty start-up fund from Iowa State
University.

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

Table 1. Application timing ratio of 11 N fertilizer types

| Fertilizer Material | Fall | Spring | At planting | After planting |
|---|---|---|---|---|
| **Dry Solid Forms** | | | | |
| Ammonium Nitrate | 0 | 0.3 | 0.35 | 0.35 |
| Ammonium Sulfate | 0.25 | 0.25 | 0.25 | 0.25 |
| Sodium Nitrate | 0 | 0.3 | 0.35 | 0.35 |
| Urea | 0.05 | 0.85 | 0.05 | 0.05 |
| Calcium Nitrate | 0 | 0.3 | 0.35 | 0.35 |



| | | | | |
|---|---|---|---|---|
| Diammonium Phosphate | 0.25 | 0.25 | 0.25 | 0.25 |
| Monoammonium Phosphate | 0.25 | 0.25 | 0.25 | 0.25 |
| Ammonium Phosphates | 0.25 | 0.25 | 0.25 | 0.25 |
| **Liquid Forms** | | | | |
| Anhydrous Ammonia | 0.6 | 0.3 | 0.05 | 0.05 |
| Aqua Ammonia | 0.6 | 0.3 | 0.05 | 0.05 |
| Nitrogen Solutions | 0.1 | 0.6 | 0.15 | 0.15 |

Ammonium Phosphates is the integration of Ammonium Phosphate compounds in different formula.


Table 2. The Chemical formula, Nitrogen content, and Nitrogen form of 11 N fertilizer types

| Fertilizer type | Chemical Formula | Nitrogen Content | Nitrogen form |
|---|---|---|---|
| Anhydrous Ammonia | $NH_3$ | 82% | $NH_4^+$-N |
| Aqua Ammonia | $NH_3$ | 22% | $NH_4^+$-N |
| Ammonium Nitrate | $NH_4NO_3$ | 34% | 50% $NH_4^+$-N, 50% $NO_3^-$-N |
| Ammonium Sulfate | $(NH_4)_2SO_4$ | 21% | $NH_4^+$-N |
| Nitrogen Solutions | $CO(NH_2)_2$, $NH_4NO_3$ | 30% | 75% $NH_4^+$-N, 25% $NO_3^-$-N |
| Sodium Nitrate | $NaNO_3$ | 16% | $NO_3^-$-N |
| Urea | $CO(NH_2)_2$ | 46% | $NH_4^+$-N |
| Calcium Nitrate | $Ca(NO_3)_2$ | 17% | $NO_3^-$-N |
| Diammonium Phosphate | $(NH_4)_2HPO_4$ | 18% | $NH_4^+$-N |
| Monoammonium Phosphate | $NH_4H_2PO_4$ | 11% | $NH_4^+$-N |
| Ammonium Phosphates | $(NH_4)_3PO_4$, etc. | 15% | $NH_4^+$-N |

Ammonium Phosphates (APs) is the integration of Ammonium Phosphate compounds in different formula. We assumed its nitrogen content is 15% through 1900 to 2015. Although one of APs is Ammonium Phosphate Nitrate (APN), we still considered APs is $NH_4^+$-N due to APN shares very small proportion of APs (<0.001).




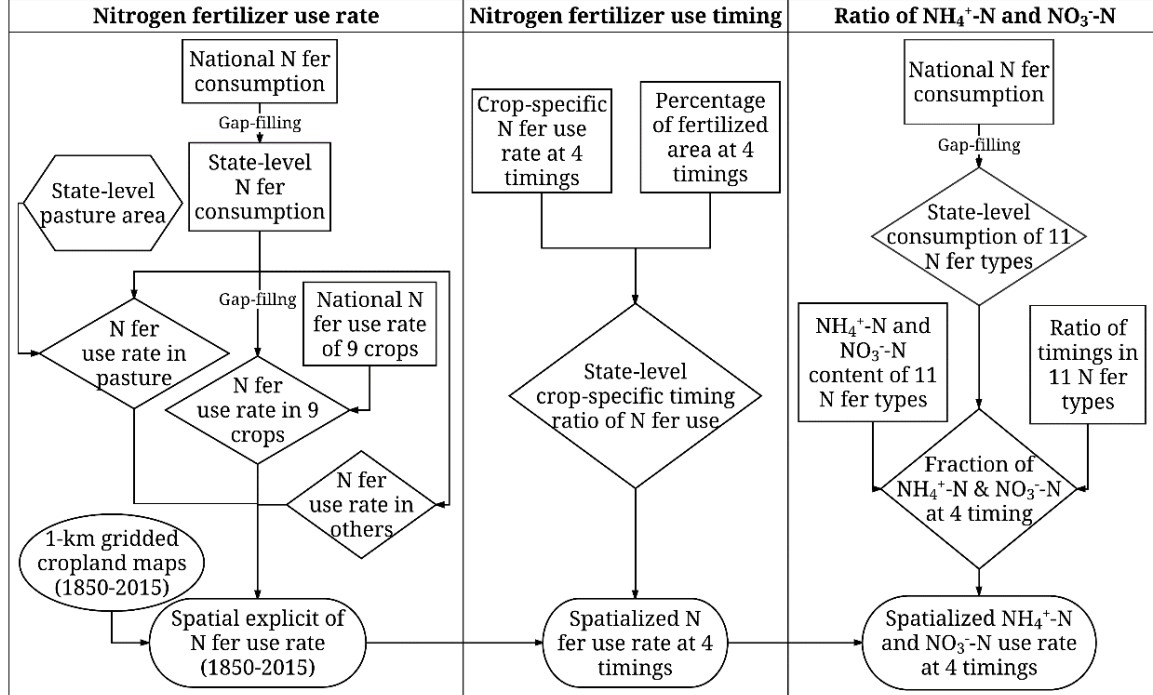

**Figure 1: Workflow diagram for developing spatially explicit time-series N fertilizer use data (including rate, timing, and NH$_4^+$-N and NO$_3^-$-N use rate) in the continental U.S. during the period of 1850-2015.**






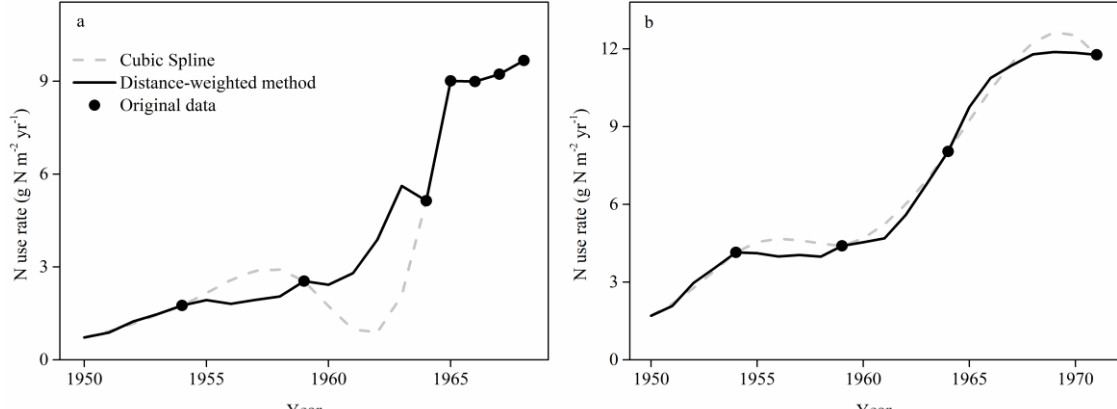

**Figure 2: Difference between distance-weighted imputation and cubic spline interpolation on large gaps (> 3 years). Grey dash line is the gap-filling by cubic spline, black line is the gap-filling by distance-weighted imputation, and dots are the original data. (a) N use rate of corn in Maryland. (b) N use rate of corn in Delaware.**






**Figure 3: Time series of N fertilizer use in the United States: (a) total commercial N fertilizer consumption and average use rate of the continental U.S. during 1940-2015 (figure inset provide a longer time series of N fertilizer use spanning from 1860 to 2015); (b) national crop-specific N fertilizer application rate (error bars indicate standard error within states in one year).**





**Figure 4: Spatial distribution of N fertilizer uses in the contiguous United States from 1900 to 2015. Values represent agricultural average N use rate over all crops in each 5 km by 5 km grid cell.**


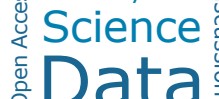



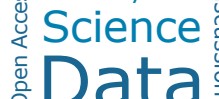

**Figure 5: Geographic distribution of N application rate in 4 application timings of the year 2015: (a) fall (previous year), (b) spring (before planting), (c) at planting, and (d) after planting, respectively.**






**Figure 6: Regional fraction of 11 N fertilizer types from 1920 to 2015. The fraction was the percentage of N content of each N fertilizers to total N consumption. 11 N fertilizers include Anhydrous Ammonia (AnA), Aqua Ammonia (AqA), Ammonium Nitrate (AN), Ammonium Sulfate (AS), Nitrogen Solution (NS), Sodium Nitrate (SN), Urea, Calcium Nitrate (CN), Diammonium Phosphate (DAP), Monoammonium Phosphate (MAP), and Ammonium Phosphates (APs).**



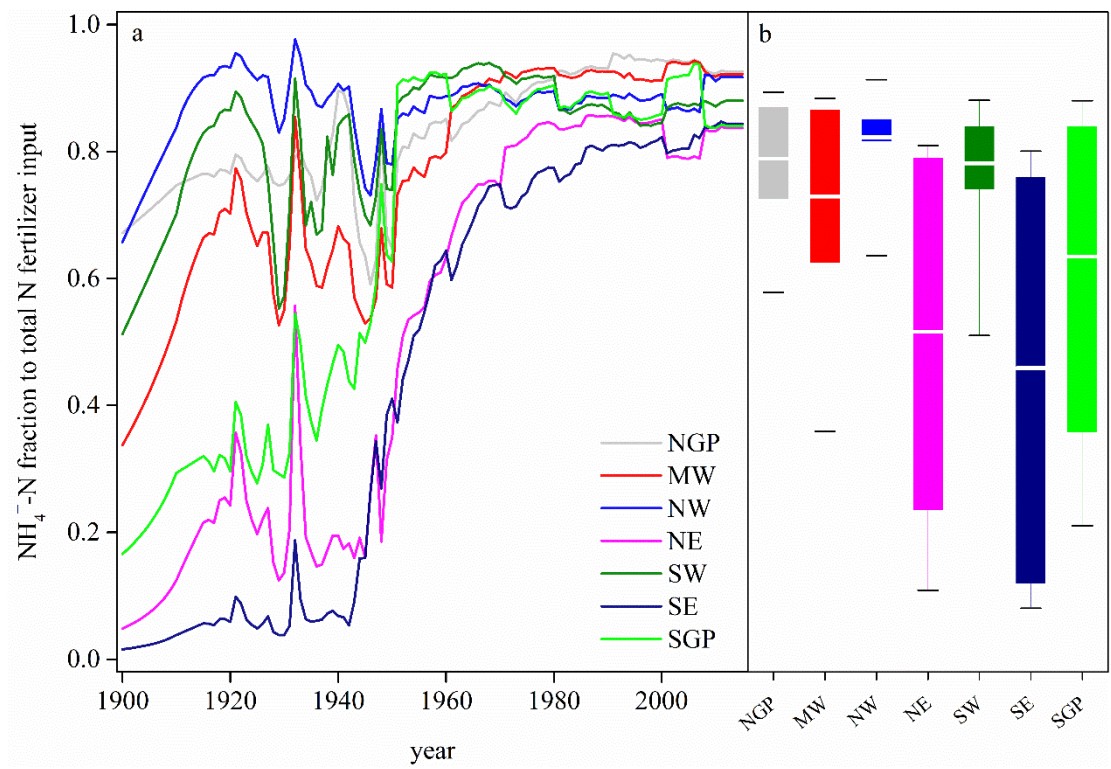

**Figure 7: Regional fraction of NH4+-N to total N use in the contiguous United States during 1900-2015. (a) Historical change of NH4+-N fraction across seven regions. (b) Boxes include 25-75% of regional fractions during 1900-2015, white lines are mean values and whiskers comprise the whole range of data. Seven regions are the Northern Great Plains (NGP), the Midwest (MW), the Northwest (NW), the Northeast (NE), the Southwest (SW), the Southeast (SE), the Southern Great Plains (SGP).**





**Figure 8: Spatial distribution of NH₄⁺-N and NO₃⁻-N application rate in 4 application timings of the year 2015.**