# Peer review of "Historical Nitrogen Fertilizer Use in Agricultural Ecosystem of the Continental United States during 1850-2015: Application rate, Timing, and Fertilizer Types"

_Earth System Science Data, 2017_

## Referee Comment (RC1) · Anonymous Referee #1 · 27 Jan 2018

Cao et al. present a comprehensive spatial dataset of historical nitrogen fertilizer use in the United States. The authors are to be commended for undertaking an ambitious project and integrating new types of data useful to the agriculture and biogeochemical modeling communities, including fertilizer type and timing of application.

While the general methodological approach makes sense, my primary critique is that the rationale for the authors' specific choices is not always clear and the methodological description is confusing in some places.

[Figure]

A major limitation is that it appears fertilizer use by crop is spatialized to a cropland map that is not crop-specific. This needs to be made clear, and the limitations of the resulting spatial patterns discussed. Does state or county cropland area enter into the analysis? This could lead to some substantial biases in the data product.

Paragraph at 115: There is a reference here to eq. 1 as providing the state N fertilizer use rate of each crop, but eq. 1 is only total N consumption in each state.

Paragraph at 118: It's unclear what benefit is provided by the cubic spline approach, since it shows some strange values for Maryland in Fig. 2. Why bother with two different approaches?

Paragraph at 135: The specification of these regression models is unclear from the text - please show examples using equations.

Overall, it should be clearer if and how the derived rates are (or are not) consistent with state and national consumption estimates.

Pages 5-6: Is the Agronomy Guide only used to populate Table 1, whereas the timing maps are coming from the survey? This could be clearer. Also, is it fair to say the Agronomy Guide is providing recommended timing or is this based on any survey data? The table legend should be more explicit.

Section 2.4 - Please provide more background about the historical land cover dataset, and its coverage for different land cover and crop types (see earlier comment).

Paragraph at line 216: Histograms across total land area or agricultural area could be used to show graphically the change in distribution.

Since many numbers are reported as rates over all land area, the results are not immediately accessible to those who may be interested in how cropland application rates have changed over time (although these numbers could be backed out of the dataset). I think the authors could demonstrate the richness of the dataset by highlighting a case study, e.g. maize in Iowa, and then show how the rates (per maize area) have changed

over time, the timing, and fertilizer type.

Finally, it would be helpful to show more comparisons with other N fertilizer maps, including global products which may rely upon coarser land use maps. The authors have their own map that should be compared against, for example. For this, it would be good to put the datasets on a common grid and also include difference maps.

Line 10: "essential implications" - awkward wording

Line 208: "consumptions" -> "consumption"

Line 349: "maximizing crop production" -> "maximizing profit" would be more accurate

---

## Referee Comment (RC2) · Anonymous Referee #2 · 8 Mar 2018

I appreciate that the editor has offered this albeit late opportunity to comment. The dataset of historical nitrogen fertilizer use in agricultural ecosystem across the Continental United States itself is already quite important, not to mention its importance to assess the key biogeochemical processes and water quality. I do understand that there is no perfect way to generate such dataset. So I think this dataset and paper is worth to publish in this journal. But there are some information needs to be clarified before its acceptance and publication.

Line 87-90. For different datasets at national level, is there any gap among those

datasets or do they have smooth connection among different datasets. For Mehring et al., 1957, do they provide annual data for each year?

For the crop pasture N consumption, did the author assume the N were evenly distributed? In addition, I was a little suspicious about using the ratio for the year 1964 to cover entire period during 1945-2015. At least, the authors should discuss the related uncertainties raised.

Regarding the temporal change of nitrogen fertilizer consumption, what is the reason for the peak and big drop during 1980s and near 2010?

There are many approximations required to produce the nitrogen fertilizer maps, and then to allocate the Nfer to different crops at different time period. The authors may need to discuss these in greater detail than you do in the "uncertainties" section.

---

## Author Comment (AC1) · 5 Apr 2018

**Response to Review 2:**

We thank the reviewer for providing the insightful comments and constructive suggestions, which greatly improve the quality of the manuscript. Please see our responses and related changes (highlighted in red) as below.

*I appreciate that the editor has offered this albeit late opportunity to comment. The dataset of historical nitrogen fertilizer use in agricultural ecosystem across the Continental United States itself is already quite important, not to mention its importance to assess the key biogeochemical processes and water quality. I do understand that there is no perfect way to generate such dataset. So I think this dataset and paper is worth to publish in this journal. But there are some information needs to be clarified before its acceptance and publication.*

*Line 87-90. For different datasets at national level, is there any gap among those datasets or do they have smooth connection among different datasets. For Mehring et al., 1957, do they provide annual data for each year?*

**Reply:** In this study, we developed three national data, including national N consumption, national average crop-specific N use rate, and national consumption of 11 N fertilizer types for different purpose. They are all same source data but cover different periods. Therefore, they are smoothly connected and have no gaps between different datasets.

Mehring et al. (1957), USDA (1971), and USDA-ERS (2013) were used to develop the national N consumption. All three datasets provide annual data.

Five datasets were used to develop the national average crop-specific N use rate. Each dataset provides the average fertilizer use rate of each crop for a certain year or period (Supplementary table S3). They are all sourced from USDA survey but with slightly different features or spanning period. For the period of 1965-2015, the nitrogen fertilizer use survey was periodically conducted by each state based on their own

schedules. In addition, the survey schedules also varied among crop types within the state. Collectively, there are many gaps with irregular length in the century-long N fertilizer use data, and most of these gaps are less than three years. In addition, Mehring et al. (1957) reported the sum of multiple fertilizer uses (N, P, K, trace nutrients) for each crop, instead of N fertilizer use, in the year 1927, 1938, 1842, and 1946. They provided data of both N fertilizer and total fertilizer use in 1950. Therefore, the crop-specific N fertilizer use in other years have been reconstructed based on the ratio of N input to the sum of multiple fertilizer uses in 1950.

Four datasets were used to reconstruct the national consumption of 11 N fertilizer types, they provide annual value for different period and are smoothly connected (Supplementary table S6).

Supplementary table S3. Data sources for national average crop-specific N use rate

| Data sources | Period | Crop types | Data form |
|---|---|---|---|
| Mehring et al. (1957) | 1927, 1938, 1942, 1946, 1950 | 7[a] | Total fertilizer consumption |
| USDA (1957) | 1954 | 10[b] | N use rate |
| Ibach et al., (1964) | 1959 | 10[b] | N use rate |
| Ibach and Adams. (1967) | 1964 | 10[b] | N use rate |
| USDA-ERS (2013) and -NASS (2017) | 1965-2015 | 9[c] | N use rate |

The number of crop types surveyed in different sources varied. [a], crop types included corn, soybeans, wheat (spring wheat, winter wheat, and durum wheat in total), cotton, sorghum, rice, barley, and cropland pasture. [b], crops included all nine major crops and cropland pasture. [c], crops only included nine major crops. Total fertilizer uses of each crop type reported in Mehring et al. (1957) contained all N, P, K, and trace fertilizers.

Supplementary table S6. Data sources used to reconstruct historical consumption record of 11 N fertilizers across nation

[Figure]

| Fertilizers | AnA | AqA | AN | AS | NS | SN | Urea | CN | DAP | MAP | APs |
|---|---|---|---|---|---|---|---|---|---|---|---|
| 1900-1953 | | | | | | | | | \ | \ | |
| 1954-1959 | | | | | | | | \ | \ | \ | |
| 1960-2003 | | | | | | | | | | | |
| 2004-2011 | | | | | | | | | | | |

Red: Mehring et al. (1957), green: USDA (1966), blue: USDA-ERS (2013), yellow: FAO (2017). AnA: Anhydrous Ammonia. AqA: Aqua Ammonia. AN: Ammonium Nitrate. AS: Ammonium Sulfate. NS: Nitrogen solution. SN: Sodium Nitrate. CN: Calcium Nitrate. DAP: Diammonium Phosphate. MAP: Monoammonium Phosphate. APs is the integration of Ammonium Phosphates, before 1960, the consumption of DP and MP were relatively small, so these two fertilizers were incorporated in APs, after 1960, however, the consumption of these two fertilizers increased to very high amount and were reported separately.

References:

FAO (Food and Agriculture Organization of the United Nations): FAO online database, available at: http://www.fao.org/faostat/en/#data/RF, last access: 19 October 2017, 2017.

Ibach, D. B. and Adams, J. R.: Fertilizer Use in the United States by Crops and Areas, 1964 Estimates, USDA-Economic Research Service and Statistical Reporting Service, Statistical Bulletin No. 408, Washington, D. C., 1967.

Ibach, D. B., Adams, J. R., and Fox, E. I.: Commercial Fertilizer Used on Crops and Pasture in the United States, 1959 Estimates, USDA-Economic Research Service and Agricultural Research Service, Statistical Bulletin No. 348, Washington, D. C., 1964.

Mehring, A. L., Adams, J. R., and Jacob, K. D.: Statistics on Fertilizers and Liming Materials in the United States, USDA-Agricultural Research Service, Statistical Bulletin No. 191, Washington, DC, 1957.

USDA-ERS (U.S. Department of Agriculture-Economic Research Service): Fertilizer Use and Price, available at: https://www.ers.usda.gov/data-products/arms-farm-financial-and-crop-production-practices/, last access: 19 November 2017, 2013.

USDA (U.S. Department of Agriculture): Fertilizer Used on Crops and Pasture in the United States, 1954 Estimates, USDA-Agricultural Research Service, Statistical Bulletin No. 216, Washington, D. C., 1957.

*For the crop pasture N consumption, did the author assume the N were evenly distributed?*

*In addition, I was a little suspicious about using the ratio for the year 1964 to cover entire period during 1945-2015. At least, the authors should discuss the related uncertainties raised.*

**Reply:** Thanks to the reviewer for raising this insightful question and we made improvements in our data. In our study before improvements, we inferred the N fertilizer use for cropland pasture of 1945-2015 based on state-level percentage of cropland pasture N consumption in state total. We used the percentage of the year 1954 for 1945-1959 and the year 1964 for 1960-2015. So the N use in cropland pasture were not evenly distributed among states.

We made improvements of replacing the fixed state-level ratio of cropland pasture in 1954 and 1964 with dynamic annual state-level ratio from 1945 to 2015.

We made the modifications in the **2.1 Historical N fertilizer use rate reconstruction, Estimation crop-specific N use rates at state-level** of the manuscript**,** and **10 pencentage of N fertilizer use in nonfarm, permanent pasture, and cropland pasture to national N fertilizer use** in the supplementary file.

Manuscript:

Page 5, line 143. We assumed cropland pasture was not fertilized until 1945 due to a lack of area data and low N use ($< 1.5\%$ of national total N use in 1942, Mehring et al., 1957). By timing the annual state-level N consumption with the ratio of cropland pasture N consumption to total N consumption in each state of the same year derived from multiple data sources (see Supplementary table S2 for details), we obtained state-level N consumption of cropland pasture from 1945 to 2015.

Supplementary:

**10 pencentage of N fertilizer use in nonfarm, permanent pasture, and cropland pasture to national N fertilizer use**

To exclude N fertilizer use in nonfarm and permanent pasture from total N fertilizer use as non-agricultural fertilizer use and introduce the N fertilizer use in cropland pasture (supplementary table S2), we integated the state-level nonfarm, permanent pasture, and cropland pasutre N fertilizer use proportion based on Mehring et al. (1957) for 1927, 1938, 1942, 1946, and 1950, USDA (1957) for 1954, Ibach et al. (1964) for 1959, Ibach and Adams (1967) for 1964, Brakebill and Grinberg (2017) for 1987-2012 (nonfarm use), and IFA (2018) for 2015 (permanent pasture and cropland pasture). According to the newest data set published in IFA (2018), permanent pasture and cropland pasture in the U.S. together accounted for 11.3% of total N fertilzier consumption. We used the individual ratio of these two pastures of 1964 to split 11.3% to 3.8% for permanent pasture and 7.5% for cropland pasture in 2015. Thus, we calculated the increase rates of these two pastures from 1964 to 2015 and increased state-level percentage with the same rate. We assumed the state-level ratio before 1927 kept the same as the ratio of 1927 and adopted linear interpolation to gap-fill the missing years from 1927 to 2015. Supplementary figure S1 shows how the national percentage of N fertilizer use in nonfarm (red line), permanent pasture (green line), cropland pasture (blue line), and nonfarm and permanent pasture together (black dashed line) changed through 1927 to 2015.

[Figure]

Supplementary Figure S1: Percentage of N fertilizer use in nonfarm, permanent pasture, and cropland pasture (classification can be found in Table S2 of Supplementary Information) in the U.S. from 1927 to 2015. Red line represents nonfarm, green line is permanent pasture, blue line is cropland pasture, and black dashed line is sum of nonfarm and permanent pasture.

*Regarding the temporal change of nitrogen fertilizer consumption, what is the reason for the peak and big drop during 1980s and near 2010?*

**Reply:** Nitrogen fertilizer use can be affected by many factors, such as land use change, agricultural management (crop rotation, fertilizer type change, etc.), technological development (genetic improvement, etc.), economic benefit (labor cost, fertilizer price, crop price, etc.), and local policies. We extended the text in **4.2 Temporal and Spatial change in nitrogen fertilizer use** of manuscript to explain such abrupt changes.

Page 10, line 310. together with the markedly increase in application rate of all crops except cotton, the major agricultural regions received tremendous amount of N during

1970 to 1985, except the conspicuous drop in the year 1983 due to the large cropland area abandon (Yu and Lu, 2017). Driven by the change of grain demand and fertilizer prices, N fertilizer use in the U.S. gradually increased with more fluctuation after 1985, such as the drop of fertilizer consumption in 2008-2009, which may be suppressed by the high price of N fertilizer caused by the 2008 financial crisis (USDA-ERS, 2013).

References:

Yu, Z. and Lu, C. Q.: Historical cropland expansion and abandonment in the continental U.S. during 1850 to 2015, Glob. Ecol. Biogeogr., 27, 322-333, 2017, DOI: 10.1111/geb.12697.

USDA-ERS (U.S. Department of Agriculture-Economic Research Service): Fertilizer Use and Price, available at: https://www.ers.usda.gov/data-products/arms-farm-financial-and-crop-production-practices/, last access: 19 November 2017, 2013.

*There are many approximations required to produce the nitrogen fertilizer maps, and then to allocate the Nfer to different crops at different time period. The authors may need to discuss these in greater detail than you do in the "uncertainties" section.*

**Reply:** We agree with the reviewer that there are unavoidable uncertainties in data processing and development. Our N fertilizer data include three final products, i.e., maps of annual N fertilizer use rate in the continental US at 1-km resolution from 1850-2015 (by total rate, at four timings, by application of $NH_4^+$-N and $NO_3^-$-N at four timings).

We have added the description about the historical land cover dataset developed by Yu and Lu, (2017) to the **2.4 Spatializing state-level crop-specific N fertilizer input to gridded maps**, and relevant uncertainties derived from this into **4.3 Uncertainty and future research needs** of **4 Discussion** in the manuscript.

Page 7, line 197. For spatial analysis, we downscaled the imputed state-level crop-specific N management data to gridded maps based on 1 km × 1 km historical land cover data (including crop density and crop type distribution maps) of the contiguous U.S. from 1850 to 2015 developed by Yu and Lu (2017) and Yu et al. (2018). The

cropland density maps, by incorporating various sources of inventory data and high spatial resolution satellite images, were reconstructed to represent the area of cropped land each year while excluding summer idle/fallow. The crop type maps were reconstructed using satellite images and the USDA National Agricultural Statistics Service (NASS) survey data, and state-level land area of each crop type in each year is consistent with USDA survey. More details about cropland maps can be found in Yu and Lu (2017) and Yu et al. (2018).

Page 12, line 362. 6) The historical crop type maps were reconstructed using USDA survey data at state-level. However, spatial distribution of N fertilizer use was uncertain in sub-state level because the lack of finer scale data for crop type map reconstruction.

References:

Yu, Z. and Lu, C. Q.: Historical cropland expansion and abandonment in the continental U.S. during 1850 to 2015, Glob. Ecol. Biogeogr., 27, 322-333, 2017, DOI: 10.1111/geb.12697.

Yu, Z., Lu, C., Cao, P., and Tian, H.: Long-term terrestrial carbon dynamics in the Midwestern United States during 1850-2015: Roles of land use and cover change and agricultural management, Glob. Change Biol., 1-18, 2018, DOI: 10.1111/gcb.14074.

---

## Author Comment (AC2) · 5 Apr 2018

**Response to Review 1:**

We thank the reviewer for scrutinizing our manuscript and providing the insightful comments and constructive suggestions, which greatly improve the quality of the manuscript. Please see our responses to the comments and related changes (highlighted in red) as follows.

*Cao et al. present a comprehensive spatial dataset of historical nitrogen fertilizer use in the United States. The authors are to be commended for undertaking an ambitious project and integrating new types of data useful to the agriculture and biogeochemical modeling communities, including fertilizer type and timing of application. While the general methodological approach makes sense, my primary critique is that the rationale for the authors' specific choices is not always clear and the methodological description is confusing in some places.*

*A major limitation is that it appears fertilizer use by crop is spatialized to a cropland map that is not crop-specific. This needs to be made clear, and the limitations of the resulting spatial patterns discussed. Does state or county cropland area enter into the analysis? This could lead to some substantial biases in the data product.*

**Reply:** Thanks to the reviewer to point out the unclear description about the cropland distribution map we used in this study. The cropland database developed by Yu and Lu (2017) includes annual cropland density maps and crop type distribution maps covering the conterminous US spanning from 1850 to 2016.

We have added the description about the historical land cover dataset developed by Yu and Lu, (2017) to the **2.4 Spatializing state-level crop-specific N fertilizer input to gridded maps** in the manuscript.

Page 7, line 197. For spatial analysis, we downscaled the imputed state-level crop-specific N management data to gridded maps based on 1 km × 1 km historical land

cover data (including crop density and crop type distribution maps) of the contiguous U.S. from 1850 to 2015 developed by Yu and Lu (2017) and Yu et al. (2018). The cropland density maps, by incorporating various sources of inventory data and high spatial resolution satellite images, were reconstructed to represent the area of cropped land each year while excluding summer idle/fallow. The crop type maps were reconstructed using satellite images and the USDA National Agricultural Statistics Service (NASS) survey data, and state-level land area of each crop type in each year is consistent with USDA survey. More details about cropland maps can be found in Yu and Lu (2017) and Yu et al. (2018).

References:

Yu, Z. and Lu, C. Q.: Historical cropland expansion and abandonment in the continental U.S. during 1850 to 2015, Glob. Ecol. Biogeogr., 27, 322-333, 2017, DOI: 10.1111/geb.12697.

Yu, Z., Lu, C., Cao, P., and Tian, H.: Long-term terrestrial carbon dynamics in the Midwestern United States during 1850-2015: Roles of land use and cover change and agricultural management, Glob. Change Biol., 1-18, 2018, DOI: 10.1111/gcb.14074.

*Paragraph at 115: There is a reference here to eq. 1 as providing the state N fertilizer use rate of each crop, but eq. 1 is only total N consumption in each state.*

**Reply:** thanks for pointing this out. In our study, equation 1 and equation 2 are not only applied in imputing national crop-specific N use rate, but also used in generating state-level crop-specific N use rate, and state-level consumption of Calcium Nitrate, Diammonium Phosphate, and Monoammonium Phosphate. To avoid the confusion, we reconstructed the equations and rephrased the sentence.

Page 4, line 117.

$$Raw\ data_i = \frac{Referenced\ trend_i}{Referenced\ trend_{i+1}} \times Raw\ data_{i+1} \tag{1}$$

where *Raw data* refers to the raw data that contains missing values, the *Referenced trend* refers to the complete data from which we extracted the changing trend that raw data can refer to, and *i* is the years between 1850 to 1930.

Page 4, line 125.

$$Raw\ data_{i+k} = \frac{Referenced\ trend_{i+k} \times Raw\ data_i}{Referenced\ trend_i} \times \frac{k-i}{j-i} + \frac{Referenced\ trend_{i+k} \times Raw\ data_j}{Referenced\ trend_j} \times \frac{j-k}{j-i}$$

(2)

where *Raw data* refers to the raw data that contains missing values, *Referenced trend* refers to the complete data that provides a reliable changing trend, the years *i* and *j* were the beginning and ending year of the gap, and the year *i+k* was the $k^{th}$ missing year.

*Paragraph at 118: It's unclear what benefit is provided by the cubic spline approach, since it shows some strange values for Maryland in Fig. 2. Why bother with two different approaches?*

**Reply:** We agree with the reviewer that it is important to stress on the benefit of adopting these two approaches under different situations. The nitrogen fertilizer use survey was periodically conducted by each state based on their own schedules. In addition, the survey schedules also varied among crop types within the state. Collectively, there are many gaps with irregular length in the century-long N fertilizer use data, and most of these gaps are less than three years. Cubic spline interpolation is advantaged in high efficiency (using "spline" command with "fmm" method in R). More importantly, cubic spline method imputes the missing values smoothly by considering the effects of points at sides of the gap based on the third-order polynomials. Nonetheless, we found that cubic spline often over-fits the N rates when the gaps are more than three years. Whereas the distance-weighted method is capable to reproduce the interannual variation of N use rates. In comparison, when the missing years are less than three years, the distance-weighted method may largely bias the gap-filled values when the trend of referenced rates is opposite to that of original Thus, we use cubic spline to gap-fill N use rates and avoid abrupt biases.

We have added the comparison of these two approaches into **11 Comparison between cubic spline method and distance_weighted method in interpolating the missing**

**values** in the supplementary file.

**11 Comparison between cubic spline method and distance-weighted method in interpolating the missing values**

We adpoted two methods, cubic spline method (for gaps less than three years) and distance-weighted method (Eq. 2, for gaps larger or equal to three years), to gap-fill the missing values in state-level crop-speicific N use rates. It is of high probility that the trend of referenced rates are reverse to the row rates in short periods (less than three years), which may produce abrupt biases while using distance-weighted method (Fig. S2a&b). whereas cubic spline could interpolate the missing values more smoothly due to it adopts the third-order polynomials to interpolate the missing values, in which the function fits through at most four points at side of the gap. In comparison, cubic spline may over-fits the N use rates and fails to capture the long-term interannual variation of the N use rates in long period (larger than three years) (Fig. S1c&d). Whereas the distance-weighted approach, which considers the contribution of the weight of the neatest value and uses the trend of corrected rate (Fig. S3c) as base-line, are capable to restore reproduce the interannual variation of N use rates.

[Figure]

Supplementary figure S2: Difference between distance-weighted imputation and cubic spline interpolation on different length of gaps. (a) One missing year gap-filling based on distance-weighted method, (b) Two missing years gap-filling based on distance-weighted method, (c) Comparison of two gap-filling methods in N use rate of corn in Maryland, (d) Comparison of two gap-filling methods in N use rate of corn in Delaware. Grey line is referenced rate, red line is N fertilizer rate gap-filled by distance-weighted imputation, blue line is N fertilizer rate gap-filled by cubic spline approach, triangle is the gap-filled value, and dots are the raw data.

*Paragraph at 135: The specification of these regression models is unclear from the text - please show examples using equations.*

**Reply:** The details of regression models and how they were applied in gap-filling state-level crop-specific N use rates have been added to the supplementary material. Please see more details in section **4 Regression models (Table S4)** and section **12 Gap-filling the raw state-level crop-specific N use rates with referenced N use rates by regression model (Fig S3)** in the supplementary file.

**4 Regression models**

The referenced state-level crop-specific N use rates generated in section 2.1 combined national crop-specific N use rates with state-level N use rates, which can potentially provide the referenced interannual variation and temporal trend for gap-filling the missing values in the raw state-level crop-specific N use rates. But since national crop-specific N use rates and state-level N use rates are the average of N used by a given crop across the nation and by all crops within a given state, their combination, here we named the referenced state-level crop-specific N use rates, mismatchs the amplitidue and changing rate of the state-level crop-specific N use rates more or less (Fig. S2a). So we built regression models (Table S4) between the referenced crop-specific rates and raw state-level crop-specific rates to tune these two data sets to the same amplitude and changing rate (Fig. S2b&c).

Supplementary table S4. Regression models used to gap-fill raw state-level crop-specific N use rates

| Model | Equation |
|---|---|
| Quadratic Model | $y = ax^2 + bx + c$ |
| Cubic Mode | $y = ax^3 + bx^2 + cx + d$ |
| Exponential Model | $y = ae^x + b$ |
| Logarithmic Model | $y = alog(x) + b$ |

Where $y$ is the raw state-level crop-speicific N use rates, $x$ is the referenced state-level crop-specific N use rates.

**12 Gap-filling the raw state-level crop-specific N use rates with referernced N use rates by regression model**

We combined the national crop-specific N use rates and state-level N use rates as the reference for gap-filling the raw data of state-level crop-specific N use rates. However, there are discrepancies in the amplitude and the changing rate between the referenced N use rates and raw N use rates (Fig S3a), which produce abrupt changes (outliers) during the gap-filling. Thus, we first built the regression models (Table S4) between referenced rates and raw rates (Fig. S3b) to make the correction on the amplitude and changing rate of referenced rates (Fig. S3c). Then we gap-filled the missing values

based on the corrected reference (Fig. S3d).

[Figure]

Supplementary figure S3. Gap-fill the missing N use rate of Spring Wheat in South Dakoda Spring wheat during 1954 to 2015. (a) Referenced rate and raw data (blue line means there is no missing years between the data points). (b) Regression model we built between raw data and referenced rate, which was used to correct referenced rate for better magnitude and changing rate (The Cubic Model is the best regression model with high $R^2$ of 0.96). (c) the referenced rate (black line) and corrected reference (gray line) based on cubic model in fig b. (d) Final product of gap-filled values (green line) by using weighted-distance approach and corrected reference.

*Overall, it should be clearer if and how the derived rates are (or are not) consistent with state and national consumption estimates.*

**Reply:** The development of state-level crop-specific nitrogen (N) use rates in this study was controlled by the state agricultural N consumption (excluding nonfarm use and permanent pasture use, supplementary table S2 and fig. S1), which means sum of state N consumption from the derived rates should theoretically equals to the state agricultural N consumption provided by the USDA. We first gap-filled the N use rates

of nine major crops (corn, soybean, winter wheat, spring wheat, cotton, sorghum, rice, barley, and durum wheat) of each state, then we derived state-level N use rates of cropland pasture from 1945 to 2015. We assume the rest of N fertilizer in each state was applied to other crops, which vary widely among states. By keeping the sum of crop-specific fertilizer use consistent to state total, we ensure our data is consistent with survey-based state and national agricultural fertilizer consumption in magnitude over time (excluding non-farm use and permanent pasture use).

*Pages 5-6: Is the Agronomy Guide only used to populate Table 1, whereas the timing maps are coming from the survey? This could be clearer. Also, is it fair to say the Agronomy Guide is providing recommended timing or is this based on any survey data? The table legend should be more explicit.*

**Reply:** Nitrogen fertilizer application timing (Table S4) and application timings for 11 N fertilizer types (Table 1 and Table S7) are different. Nitrogen fertilizer application timing is state-level crop-specific timing information, which is based on the survey conducted by USDA-ERS. This timing information was used for splitting annual state-level crop-specific N use rates into four timings (Timing maps, Fig. 5), but there is no information for $NH_4^+$-N/$NO_3^-$-N ratio. The application timing for 11 N fertilizer types was derived from the Agronomy Guide and a survey in Minnesota (Bierman et al., 2012). It provides the reference to split annual consumption of each N fertilizer type (e.g. Anhydrous Ammonia, Ammonium Nitrate, Urea, etc.) into four timings. We adopted it to further divide state-level crop-specific N use at each timing to $NH_4^+$-N and $NO_3^-$-N use rates. Such information can be found in the methodology and the workflow chart in the manuscript. Here we provide a flowchart to help illustrate these processes.

[Figure]

*Section 2.4 - Please provide more background about the historical land cover dataset, and its coverage for different land cover and crop types (see earlier comment).*

**Reply:** We have added the description about the historical land cover dataset developed by Yu and Lu, (2017) to the **2.4 Spatializing state-level crop-specific N fertilizer input to gridded maps** in the manuscript. Please see the response to earlier comment.

*Paragraph at line 216: Histograms across total land area or agricultural area could be used to show graphically the change in distribution.*

**Reply:** We agree with the reviewer's suggestion for showing the N fertilizer change in distribution. The national N fertilizer use rates of the continental U.S. has been continuously increasing since 1850. This general increase, however, may vary among different regions due to the difference in crop cultivation. Thus, we based on the N fertilizer use rates (per land area) of 1960, 1980, and 2015 to discuss the change of the rates distribution across the continental U.S. We didn't include the year 2000 because the rates increased very slowly after the 2000s. We counted the numbers of grid cells at every 0.1 g N m$^{-2}$ yr$^{-2}$ interval from 0 to 21 g N m$^{-2}$ yr$^{-1}$ and divided the numbers by total grid cells to generate the frequency in percentage. We extended the text in **3.1**

**Nitrogen fertilizer application rate**.

Page 7, line 210. In 1960, most agricultural area received low N fertilizer input (less than 6 g N $m^{-2}$ $yr^{-1}$), while due to expansion of cropland and rapid increase in N fertilizer use from 1960 to 1980, the area receiving larger N input (2-8 g N $m^{-2}$ $yr^{-1}$) greatly increased in 1980, and frequency peaked at a few high N input values, which may be caused by intensive fertilizer use in certain crop types (e.g., maize and rice). In 2015, we found less areas were characterized by low to medium-level fertilizer input (< 8 g N $m^{-2}$ $yr^{-1}$), and high-level N fertilizer application (8-18 g N $m^{-2}$ $yr^{-1}$) was more widespread compared to 1960 and 1980 (insert figure in Fig. 2a).

[Figure]

Figure 2: Time series of N fertilizer use in the United States: (a) national commercial N fertilizer consumption and use rate of the continental U.S. during 1850-2015 derived from this study (figure inset provides the N fertilizer rates distribution across the continental U.S. in the year 1960, 1980, and 2015); (b) national crop-specific N fertilizer application rate (error bars indicate standard error across states in one year).

*Since many numbers are reported as rates over all land area, the results are not immediately accessible to those who may be interested in how cropland application rates have changed over time (although these numbers could be backed out of the dataset). I think the authors could demonstrate the richness of the dataset by highlighting a case study, e.g. maize in Iowa, and then show how the rates (per maize area) have changed.*

**Reply:** We appreciate the suggestion for highlighting a case study of rate change of a crop within state. However, our study aims at spatializing the state-level crop-specific survey data of N fertilizer use rates (per crop area) based on cropland area and distribution maps. The raw data can be downloaded from USDA-ERS (https://www.ers.usda.gov/data-products/fertilizer-use-and-price.aspx), from which the changes of fertilizer uses can be directly estimated. We also agree there is large spatial heterogeneity in N fertilizer use rate even for one crop type in one state, but sub-state crop-specific data is unavailable so far. We expect to have more information in the future, to improve our data set and to inform modeling assessment.

*Finally, it would be helpful to show more comparisons with other N fertilizer maps, including global products which may rely upon coarser land use maps. The authors have their own map that should be compared against, for example. For this, it would be good to put the datasets on a common grid and also include difference maps.*

**Reply:** We appreciate the reviewer's constructive suggestion in improving the manuscript. Three more N fertilizer use data sets have been added to compare with our data set. One is county-level N fertilizer use in the continental U.S. developed by IPNI (NuGIS), another two are global N fertilizer use derived from IFA and FAO, respectively.

We have add the comparison of other three datasets to our data in **4.1 Comparison**

**with existing N fertilizer maps** in the manuscript.

Page 9, line 310. In order to examine the validity of our data, we compared our product with other studies. The temporal variations and spatial patterns of N fertilizer use were compared among our study, one regional data set from IPNI (2018), and two global data sets from Lu and Tian (2017) and Nishina et al. (2017) (Fig. 8 and Fig. 9). IPNI (2018) developed the N fertilizer use data set of the U.S. spanning from 1987 to 2012 by adopting method in Ruddy et al. (2006). It is based on county-level N consumption data derived from state-level N fertilizer sale and agricultural land acreage. Lu and Tian (2017) and Nishana et al. (2017) derived the N fertilizer use data from the International Fertilizer Association (IFA) and the Food and Agriculture Organization of the United Nations (FAO), respectively. Four data sets showed the same inter-annual variation of national N fertilizer consumption pattern from 1961 to 2014 but small difference in magnitude (Fig. 8). The N fertilizer consumption in our study is a little smaller than the value from IPNI (2018) except the year 2001 because our study excluded the permanent pasture. The historical N fertilizer use amount in our study is slightly lower than Lu and Tian (2017). This is because the N fertilizer data from Lu and Tian (2017) contains permanent pasture and nonfarm N fertilizer use and also covers territories of the U.S. rather than continental US. Whereas our data is marginally higher than Nishina et al. (2017) in most years. The overall spatial pattern of N fertilizer use rate in the continental U.S. in 2010 from our study is similar to other three maps (Fig. 9), indicating the hotspots of N fertilizer use occurred in the Midwest, West and East Coast, and some parts of the Northwest, the Southern Great Plains. Nevertheless, discrepancies were also found in the magnitude of N use rates between our data and the other studies. There were only a few states in Lu and Tian (2017) and no state in Nishina et al. (2017) applied over 7 g N m$^{-2}$ yr$^{-1}$ in the year 2010 (Fig. 9c&d). In comparison, two global data sets showed many regions received fertilizer at 2-4 g N m$^{-2}$ yr$^{-1}$, in which very little N fertilizer application was indicated in this study and IPNI (2018). It is because the global cropland maps used in these two global N fertilizer data sets were based on HYDE 3.2 (Klein Gildewijk, 2006) in Lu and Tian (2017) and HYDE 3.1 (Hurrt et al., 2011) in Nishina et al. (2017), respectively. The HYDE data overestimated cropland percentage

in low crop-coverage states while underestimating crop percentage in the US Corn Belt (Yu and Lu, 2017). This explains the different spatial patterns revealed by these four N fertilizer data sets that although their total consumptions are very close nationally. By comparison, our data considers cross-crop divergence in N fertilizer use, covers longer period, and shows more details with finer resolution.

[Figure]

Figure 8: Time series of N fertilizer use in the United States: national commercial N fertilizer consumption of the U.S. during 1961-2014 derived from four data sets. Red line: this study, Green line: Lu and Tian, (2017), Blue line: Nishina et al. (2017), and ×: IPNI (2018). The N fertilizer consumption derived from Lu and Tian (2017) covers all states in the U.S., whereas other three data sets only cover the continental U.S.

[Figure]

N fertilizer use rate (g N m⁻² yr⁻¹)    0   < 1   1 - 2   2 - 4   4 - 7   7 - 10   10 - 14   > 14

Figure 9: Comparison of N fertilizer use rate across the continental U.S. in 2010. (a) This study; (b) IPNI (2018); (c) Lu and Tian (2017); (d) Nishana et al. (2017).

References:

Hurtt, G., Chini, L. P., Frolking, S., Betts, R., Feddema, J., Fischer, G., Fisk, J., Hibbard, K., Houghton, R., Janetos, A., Jones, C. D., Kindermann, G., Kinoshita, T., Klein Goldewijk, K., Riahi, K., Shevliakova, E., Smith, S., Stehfest, E., Thomson, A., Thornton, P., van Vuuren, D. P., Wang, Y. P.: Harmonization of land-use scenarios for the period 1500–2100: 600 years of global gridded annual land-use transitions, wood harvest, and resulting secondary lands, Climatic Change, 109, 117–161, 2011.

IPNI (International Plant Nutrition Institute): A Nutrient Use Information System (NuGIS) for the U.S. Norcross, GA. November 1, 2011, available at: www.ipni.net/nugis, last access: 1 Marth 2018, 2018.

Klein Goldewijk, K.: A historical land use data set for the Holocene; HYDE 3.2, DANS, doi:10.17026/dans-znk-cfy3, 2016.

Lu, C., and Tian, H.: Global nitrogen and phosphorus fertilizer use for agriculture production in the past half century: shifted hot spots and nutrient imbalance, Earth Syst. Sci. Data, 9, 181-192, https://doi.org/10.5194/essd-9-181-2017, 2017.

Nishina, K., Ito, A., Hanasaki, N., and Hayashi, S.: Reconstruction of spatially detailed global map of NH4⁺ and NO3⁻ application in synthetic nitrogen fertilizer, Earth Syst. Sci. Data, 9, 149-162, https://doi.org/10.5194/essd-9-149-2017, 2017.

*Line 10: "essential implications" - awkward wording*

*Line 208: "consumptions" -> "consumption"*

*Line 349: "maximizing crop production" -> "maximizing profit" would be more accurate*

**Reply:** We thank the reviewer for these words correction and have corrected them.